# DATA-FREE KNOWLEDGE EXCHANGE FOR AGGREGATION-FREE HETEROGENEOUS FEDERATED LEARNING

## ABSTRACT

Heterogeneous Federated Learning (HFL) is a decentralized machine learning paradigm that enables participants to leverage distributed knowledge from diversified environments while safeguarding individual privacy. Recent works that address both data and model heterogeneity still require aggregating model parameters, which restricts architectural flexibility. Knowledge Distillation (KD) has been adopted in HFL to circumvent direct model aggregation by aggregating knowledge, but it depends on a public dataset and may incur information loss when redistributing knowledge from the global model. We propose **Federated Knowledge Exchange** (FKE), an aggregation-free FL paradigm in which each client acts as both teacher and student, exchanging knowledge directly with peers and removing the need for a global model. To remove reliance on public data, we attach a lightweight embedding decoder that produces transfer data, forming the **Data-Free Federated Knowledge Exchange** (DFFKE) framework. Extensive experiments show that DFFKE surpasses nine state-of-the-art HFL baselines by up to **18.14%**. *Anonymous Repo: https://anonymous.4open.science/r/DFFKE-0E0B.*

## 1 INTRODUCTION

As data volumes surge and privacy regulations tighten, data sharing between collaborating entities for joint learning becomes untenable. Federated Learning (FL) has emerged as a crucial framework for decentralized machine learning, enabling participants to leverage distributed data while safeguarding individual privacy. This approach has been increasingly adopted in diverse real-world applications, including financial crime detection Suzumura et al. (2022); Liu et al. (2023), medical institution collaboration Joshi et al. (2022); van de Sande et al. (2021), and closed-loop supply chain decision-making Zheng et al. (2023); Islam et al. (2023).

Traditional FL utilizes methods like FedAvg McMahan et al. (2017) to aggregate local model updates into a global model (fig. 1a), enabling collaborative training across diverse clients without necessitating data sharing. Although effective under homogeneous settings,

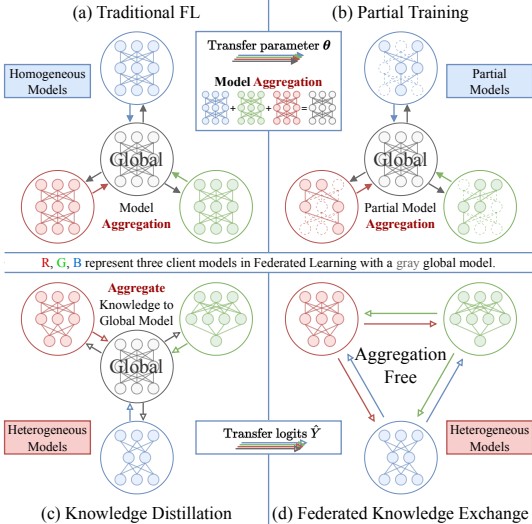

Figure 1: Comparison of three general approaches in Federated Learning with FKE.

FedAvg performance quickly declines as **data heterogeneity** increases, which is common in real-world scenarios where client data are not identically and independently distributed (Non-IID) Hsu et al. (2019). To mitigate this, subsequent FL strategies Li et al. (2020); Karimireddy et al. (2020); Acar et al. (2021); Li et al. (2021); Kim et al. (2022); Mendieta et al. (2022); Lee et al. (2022); Zhang et al. (2022a); Luo et al. (2023) modify FedAvg, incorporating mathematical constraints into the learning objectives to align local models with a global optimization goal. Distinctively, Fed-Gen Zhu et al. (2021) incorporates Data-Free Knowledge Distillation (DFKD) into FL, utilizing a

lightweight generator to produce synthetic embeddings that aid in aligning client training objectives. Subsequently, FedFTG Zhang et al. (2022b) integrates DFKD as an extension to fine-tune aggregated global models. Despite advances in addressing data heterogeneity, these approaches assume model homogeneity, relying on a uniform model architecture across clients to perform **model aggregation**. However, in practical settings, clients often vary in computational resources and may employ proprietary model architectures, resulting in significant **model heterogeneity** without prior knowledge of other clients' model choices. *This diversity limits the practicality of direct model aggregation and the redistribution of a global model to all clients.*

To tackle the challenge of model heterogeneity, recent studies have explored two main strategies: partial training (PT) and knowledge distillation (KD). PT-based approaches Caldas et al. (2018); Diao et al. (2020); Horvath et al. (2021); Alam et al. (2022); Wang et al. (2024) adapt to model heterogeneity by distributing width-based sub-models, tailored to each client's computational capacity from a large global model (fig. 1b). These sub-models are trained locally and then aggregated to enhance the global model. This method extends the FedAvg framework to accommodate resource limitations across clients, but it still restricts client model selection. Meanwhile, it also struggles with heterogeneous data because of the limitations of direct model aggregation, potentially diminishing the effectiveness and applicability of FL systems in heterogeneous environments. Recently, DFRD Wang et al. (2024) integrated DFKD within PT-based approaches to mitigate the adverse effect of heterogeneous data, addressing both data and model heterogeneity without relying on additional public datasets. Despite its promise, DFRD continues to face challenges in accommodating a broader range of client architectures, constrained by the fundamentals of model aggregation. To handle strict data and model heterogeneity while avoiding the drawbacks of model aggregation, knowledge distillation is a promising alternative. KD-based methods Li & Wang (2019); Lin et al. (2020); He et al. (2020); Afonin & Karimireddy (2021); Cho et al. (2022); Fang & Ye (2022) **aggregate knowledge** from diverse architectures by aligning the logit outputs between client models and a global model using a *public dataset* (fig. 1c). Indeed, these KD-based methods can seamlessly address both data and model heterogeneity. *Nevertheless, the effectiveness of KD relies heavily on the availability and quality of the public dataset; aggregating knowledge into a global model may inevitably lead to knowledge loss during redistribution.* Both model and knowledge aggregation impose undesirable limitations on the algorithm. Ultimately, adopting an **aggregation-free** FL paradigm is the key to overcoming this bottleneck. We present a comprehensive comparison of existing KD related approaches in table 1.

| Methods | Public Data Dependency | Require Aggregation | Support Model Heterogeneity |
|---------|----------------------|--------------------|-----------------------------|
| KD-Based | dependent | Yes / knowledge | **Yes** |
| FedGen | **data-free** | Yes / FedAvg | No |
| FedFTG | **data-free** | Yes / FedAvg | No |
| DFRD | **data-free** | Yes / PT-based | *Limited* |
| DFFKE | **data-free** | **aggregation-free** | **Yes** |

Table 1: Comparison between existing KD-based and DFKD-based Zhu et al. (2021); Zhang et al. (2022b); Wang et al. (2024) federated learning approaches with DFFKE.

In this paper, we propose *Data-Free Federated Knowledge Exchange* (DFFKE) to address the dual challenges of strict data and model heterogeneity in FL. Specifically, we propose an ***aggregation-free*** paradigm named ***Federated Knowledge Exchange*** (FKE) to facilitate multi-client knowledge exchange (fig. 1d), wherein each participant simultaneously functions as both teacher and student. This innovative dual role promotes direct knowledge sharing among clients, eliminating the need to aggregate knowledge into a global model and then redistribute it, which may cause knowledge loss during aggregation. Finally, we employ a lightweight decoder to produce synthetic transfer data to bridge the communication between client models, thereby removing dependence on public datasets and enabling ***data-free*** FKE. The main contributions of this work are summarized as follows:

- We propose FKE, a novel aggregation-free learning paradigm for heterogeneous FL, enabling direct knowledge sharing between clients and eliminating the need for a global model.

- We introduce DFFKE framework; to the best of our knowledge, it is the first FL approach that addresses both data and model heterogeneity without reliance on public datasets and enables direct client-to-client communication, which is achieved through an ***aggregation-free*** training.

- Extensive experimental results demonstrate that DFFKE significantly outperforms existing FL approaches in heterogeneous environments, showcasing its effectiveness and robustness.

## 2 NOTATIONS AND PRELIMINARIES

**Notations.** We consider a heterogeneous federated learning setting for general supervised multi-class classification tasks. Let $\mathbb{C}$ denote the set of participating clients, with $|\mathbb{C}| = K$. Each client $c_k \in \mathbb{C}$ possesses a private dataset $\mathcal{D}_k = (X_k, Y_k)$, where $X_k = \{x_i^k\}_{i=1}^{N_k} \subset \mathbb{R}^d$ is the set of data samples, and $Y_k = \{y_i^k\}_{i=1}^{N_k} \subset \mathbb{R}$ is the corresponding set of ground truth labels. Each client owns a local model $\boldsymbol{\theta}_k := [\boldsymbol{\theta}_k^h, \boldsymbol{\theta}_k^l]$, which consists of two components: a data encoder $h(\cdot) : \mathbb{R}^d \to \mathbb{R}^h$ parameterized by $\boldsymbol{\theta}_k^h$, where $h \ll d$, and a classifier $l(\cdot) : \mathbb{R}^h \to \mathbb{R}^n$ parameterized by $\boldsymbol{\theta}_k^l$. For simplicity, we denote the full network as $f(\cdot) : \mathbb{R}^d \to \mathbb{R}^n$, where $f(x_i^k; \boldsymbol{\theta}_k) = l(h(x_i^k; \boldsymbol{\theta}_k^h); \boldsymbol{\theta}_k^l)$. We also denote $\mathcal{E}_k = h(X_k; \boldsymbol{\theta}_k^h)$ and $\hat{Y}_k = f(X_k; \boldsymbol{\theta}_k)$ as the collection of embeddings and logits of client $k$, respectively. Lastly, we use $\{\boldsymbol{\theta}_k\}$ as an abbreviation for $\{\boldsymbol{\theta}_k\}_{k=1}^K$ to represent a set of all clients' model (the same for $\{\mathcal{E}_k\}, \{\hat{Y}_k\}$).

**Federated Learning (FL)** is a distributed machine learning paradigm in which data-constrained clients collaboratively train models by leveraging collective knowledge without sharing their private data. In a common FL paradigm, each client $k$ locally optimizes $\boldsymbol{\theta}_k$ on its private dataset $\mathcal{D}_k$ and then repeatedly communicates with other clients using a shared protocol. Under **model homogeneity** setting, most approaches follow the prevalent FedAvg framework McMahan et al. (2017), which aggregates the local models $\boldsymbol{\theta}_k$ into a global model $\boldsymbol{\theta}_{\text{global}}$ and then distributes it back to the clients in each communication round:

$$\boldsymbol{\theta}_{\text{global}} = \frac{1}{K} \sum_{k=1}^K \boldsymbol{\theta}_k \tag{1}$$

**Knowledge Distillation (KD)** has been proposed to transfer knowledge from a well-trained large model (teacher) to a smaller model (student) for model compression while maintaining similar performance. Traditional KD requires manually collecting public data to form a transfer dataset $\hat{\mathcal{D}}_{\text{P}} = \{\hat{x}_i^{\text{P}}\}_{i=1}^{N_{\text{P}}}$ that bridges the communication between models. The knowledge transfer is often accomplished by minimizing the Kullback-Leibler (KL) divergence Hinton (2015) between the logits produced by the teacher model $\boldsymbol{\theta}_T$ and the student model $\boldsymbol{\theta}_S$ on $\hat{\mathcal{D}}_{\text{P}}$:

$$\min_{\boldsymbol{\theta}_S} \mathbb{E}_{x \sim \hat{\mathcal{D}}_{\text{P}}} \left[ D_{\text{KL}} \left[ f(x; \boldsymbol{\theta}_T) \parallel f(x; \boldsymbol{\theta}_S) \right] \right] \tag{2}$$

**Data-Free Knowledge Distillation (DFKD)** emerges as an alternative to KD when an appropriate public dataset is unavailable. DFKD methods Chen et al. (2019); Micaelli & Storkey (2019); Choi et al. (2020); Yin et al. (2020); Fang et al. (2021; 2022); Liu et al. (2024) generate a synthetic transfer dataset $\hat{\mathcal{D}}_{\text{syn}} = \{\hat{x}_i^{\text{syn}}\}_{i=1}^n$ by extracting knowledge from a pretrained teacher model and use it to transfer knowledge by minimizing equation 2. The prevailing approach for generating $\hat{\mathcal{D}}_{\text{syn}}$ involves training a generator model $Gen$ that produces synthetic data $\hat{x}$ conditioned on a given class $y$. To ensure that $\hat{x}^{\text{syn}} = Gen(y)$ approximates the true data distribution of $y$, the generator $Gen$ minimizes the *fidelity loss*:

$$\mathcal{L}_{\text{fid}} = \sum_{y \in Y} CE\left(f(Gen(y); \boldsymbol{\theta}_T), y\right) \tag{3}$$

where *CE* denotes the cross-entropy function. Moreover, to enhance the transferability of the synthetic data $\hat{x}^{\text{syn}}$, an additional *model discrepancy loss* is introduced to encourage $\hat{x}^{\text{syn}}$ to maximize the knowledge gap (KL divergence) between the teacher model $\boldsymbol{\theta}_T$ and the student model $\boldsymbol{\theta}_S$:

$$\mathcal{L}_{\text{md}} = \sum_{y \in Y} -D_{\text{KL}} \left[ f(Gen(y); \boldsymbol{\theta}_T) \parallel f(Gen(y); \boldsymbol{\theta}_S) \right] \tag{4}$$

The overall training objective of $Gen$ in DFKD is a combination of the above losses weighted by coefficients $\alpha, \beta$:

$$\mathcal{L}_{\text{gen}} = \alpha \mathcal{L}_{\text{fid}} + \beta \mathcal{L}_{\text{md}} \tag{5}$$

## 3 DATA-FREE FEDERATED KNOWLEDGE EXCHANGE

**Knowledge Exchange.** Unlike traditional knowledge distillation (KD), which focuses on one-way knowledge transfer from a teacher model to a student model, we define Knowledge Exchange

(KE) as a learning paradigm that involves *concurrent bidirectional knowledge transfer* between models. Let $\boldsymbol{\theta}_A, \boldsymbol{\theta}_B$ represent two independent models, and let $\hat{\mathcal{D}}_A, \hat{\mathcal{D}}_B$ denote their respective transfer datasets for KE. The objective of KE is to obtain a set of knowledge-exchanged models $\tilde{\boldsymbol{\theta}}_A, \tilde{\boldsymbol{\theta}}_B$, that jointly minimize their knowledge gaps with respect to each other:

$$\min_{\tilde{\boldsymbol{\theta}}_A} \mathbb{E}_{x \sim \hat{\mathcal{D}}_B} \left[ D_{\mathrm{KL}} \left[ f(x; \boldsymbol{\theta}_B) \parallel f(x; \tilde{\boldsymbol{\theta}}_A) \right] \right] \tag{6}$$

$$\min_{\tilde{\boldsymbol{\theta}}_B} \mathbb{E}_{x \sim \hat{\mathcal{D}}_A} \left[ D_{\mathrm{KL}} \left[ f(x; \boldsymbol{\theta}_A) \parallel f(x; \tilde{\boldsymbol{\theta}}_B) \right] \right] \tag{7}$$

**Federated Knowledge Exchange.** Extending the concept of KE to a multi-model setting, we propose *Federated Knowledge Exchange* (FKE), where a group of models collaboratively exchange knowledge in a federated environment without sharing private data. Assuming $K$ clients are collaborating, FKE aims to obtain a knowledge-exchanged model $\tilde{\boldsymbol{\theta}}_k$ for each client $k$ by minimizing its knowledge gap (measured by KL divergence) relative to every other client:

$$\min_{\tilde{\boldsymbol{\theta}}_k} \sum_{i=1, i \neq k}^{K} D_{\mathrm{KL}} \left[ f(\hat{\mathcal{D}}_i; \boldsymbol{\theta}_i) \parallel f(\hat{\mathcal{D}}_i; \tilde{\boldsymbol{\theta}}_k) \right], \quad \forall k \in K \tag{8}$$

where $\hat{\mathcal{D}}_i$ denotes the transfer dataset assigned to client $i$, and $f(\hat{\mathcal{D}}_i; \boldsymbol{\theta}_i)$ represents the knowledge distribution of $\boldsymbol{\theta}_i$ on $\hat{\mathcal{D}}_i$. *In practice, assembling a public transfer dataset for each client is impractical due to data scarcity and the high overhead of data curation.*

In the following section, we introduce *Data-Free Federated Knowledge Exchange* (DFFKE), a framework that eliminates the need for public data in FKE. DFFKE operates in three key steps during each communication round. First (§3.1), we translate each client's independently evolved model embedding space into a unified embedding space to ensure alignment across clients. Next (§3.2), we train an embedding decoder model $Dec(\cdot) : \mathbb{R}^h \to \mathbb{R}^d$ that maps the unified embedding space to the data space. By using the synthetic data produced by the embedding decoder as a bridge for communication, we effectively eliminate the need for public data. Finally (§3.3), we perform federated knowledge exchange and introduce a memory buffer to facilitate efficient and effective knowledge sharing among clients. Additionally, to secure the privacy of embeddings, clients can opt to use differential privacy before sharing embeddings (§4.3). An overview of the DFFKE learning procedure is illustrated in Fig.2.

## 3.1 EMBEDDING SPACE UNIFICATION

In heterogeneous federated learning environments, clients individually train their models on private datasets, resulting in distinct embedding spaces due to independent evolution:

$$\min_{\boldsymbol{\theta}_k} \mathbb{E}_{(x,y) \sim \mathcal{D}_k} \left[ CE \left( f(x; \boldsymbol{\theta}_k), y \right) \right], \quad \forall k \in K \tag{9}$$

Such divergence in embedding space limits the possibility of training a single embedding decoder $Dec(\cdot) : \mathbb{R}^h \to \mathbb{R}^d$ for all clients. To enable effective feature utilization, it is crucial to map each clients' embeddings into a unified embedding space. We achieve this through an *embedding translation* mechanism.

For each client $k$, we introduce a pluggable *docking layer* $z(\cdot) : \mathbb{R}^h \to \mathbb{R}^h$ parameterized by $\boldsymbol{\theta}_k^z$, which is a learnable linear transformation that projects the client's embedding outputs into a shared embedding space. Let $\mathcal{E}_k = h(X_k; \boldsymbol{\theta}_k^h) = \{\varepsilon_i^k\}_{i=1}^{N_k}$ denote the set of embeddings encoded from the private data of client $k$. The docking layer transforms the embeddings as follows:

$$\tilde{\mathcal{E}}_k = z(\mathcal{E}_k; \boldsymbol{\theta}_k^z) \tag{10}$$

To align the translated embeddings of all clients, we introduce a linear global classification layer $\boldsymbol{\theta}_{\mathrm{global}}^l$ that operates on all translated embeddings $\{\tilde{\mathcal{E}}_k\}_{k=1}^K$. Recall that each embedding $\varepsilon_i^k \in \mathcal{E}_k$ has a corresponding class label $y_i^k \in Y_k$. We jointly train the docking layers $\{\boldsymbol{\theta}_k^z\}_{k=1}^K$ and the classification layer $\boldsymbol{\theta}_{\mathrm{global}}^l$ by minimizing the cross entropy:

$$\min_{\{\boldsymbol{\theta}_k^z\}, \boldsymbol{\theta}_{\mathrm{global}}^l} \sum_{k=1}^{K} CE \left( l(\tilde{\mathcal{E}}_k; \boldsymbol{\theta}_{\mathrm{global}}^l), Y_k \right) \tag{11}$$

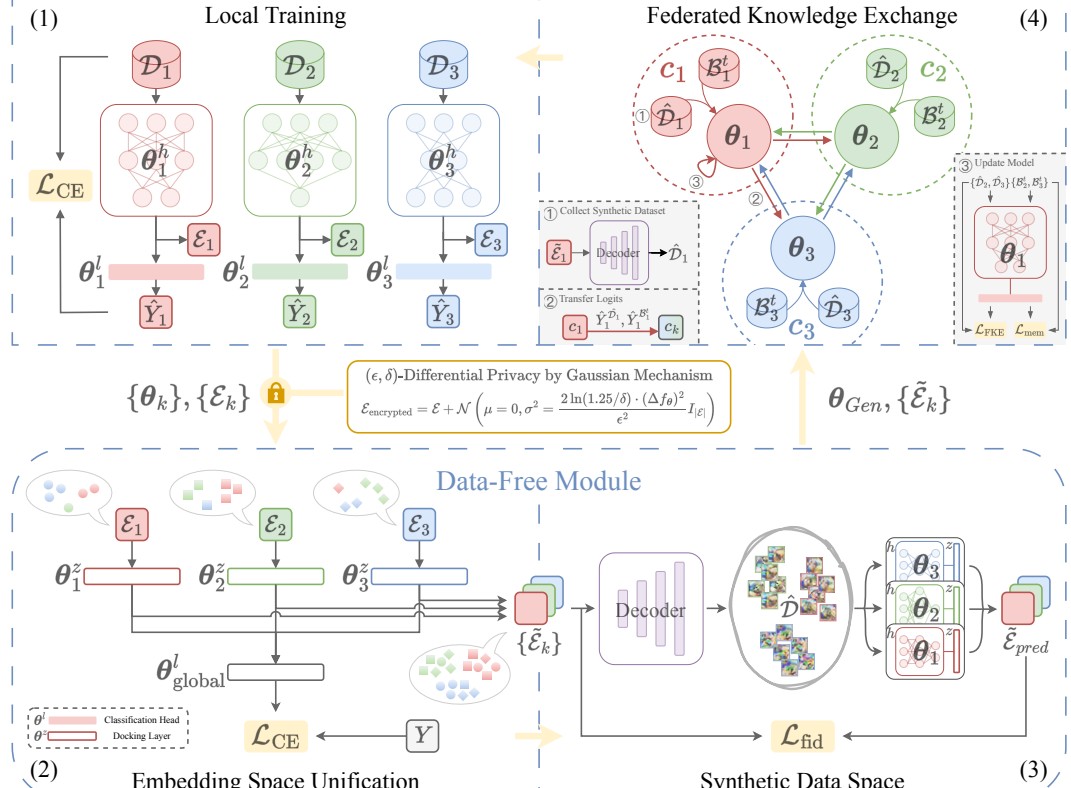

Figure 2: DFFKE comprises four procedures per communication round: (1) Training on each client's private dataset to prepare for knowledge sharing, (2) Translating each client's embeddings distribution into a unified embedding space, (3) Training an emb-decoder to map the unified embedding space to the data space, and (4) Conducting FKE using synthetic transfer data and a memory buffer.

This optimization encourages the docking layers $\theta_k^z$ to map divergent embeddings into a common embedding space where the global classifier $l(\cdot; \theta_{\text{global}}^l)$ can effectively group similar embeddings. This process can also be seen as a clustering method in which embeddings that belong to the same class from different clients are grouped together in the shared space. To fully protect client's privacy, clients can apply a differential privacy mechanism to protect their embeddings before sharing. Without loss of generality, we focus here on our FKE design and defer the detailed discussion of privacy protection to section 4.3.

## 3.2 SYNTHETIC DATA SPACE

FKE is a promising approach, but its reliance on a public dataset limits its practicality. To support communication among client models in a data-free manner, we introduce an embedding decoder $Dec(\cdot) : \mathbb{R}^h \to \mathbb{R}^d$, which maps embeddings from a unified embedding space to the data space and facilitates knowledge flow through the synthetic dataset. Moreover, unlike traditional class-guided generator methods in previous data-free approaches Zhang et al. (2022b); Wang et al. (2024), which struggle with data diversity, our embedding decoder network effectively diversifies outputs by producing distinct data based on unique embeddings. Specifically, with each client $k$'s model weight $\theta_k$, we train $Dec$ to produce a synthetic dataset $\{\hat{\mathcal{D}}_k\}$ from the input embeddings $\{\tilde{\mathcal{E}}_k\}$ by minimizing the following data fidelity loss.

**Data Fidelity** is the basis of training the generative model. $Dec$ is expected to synthesize a synthetic dataset that approximates the embedding distribution in the unified embedding space. Specifically, when synthetic data $\hat{\mathcal{D}}_k = Dec(\tilde{\mathcal{E}}_k)$ is passed through the client's encoder and docking layer, the predicted embedding $\tilde{\mathcal{E}}_{pred}$ should match the input embedding $\tilde{\mathcal{E}}_k$. This is formulated as:

$$\mathcal{L}_{\text{fid}} = \sum_{k=1}^{K} MSE\left(z(h(Dec(\tilde{\mathcal{E}}_k); \theta_k^h); \theta_k^z), \tilde{\mathcal{E}}_k\right) \tag{12}$$

By optimizing $\mathcal{L}_{\text{fid}}$, the embedding decoder learns to produce a diversified synthetic dataset that unifies the knowledge distributed across all clients. The synthetic transfer datasets $\{\hat{\mathcal{D}}_k\}$ designated to each client serve as a crucial medium for FKE.

## 3.3 KNOWLEDGE EXCHANGE WITH MEMORY BUFFER

With the trained embedding decoder $Dec$ and the unified embedding set $\{\tilde{\mathcal{E}}_k\}_{k=1}^K$ distributed to all clients, we proceed to the final Federated Knowledge Exchange (FKE) step using synthetic data. Each client retrieve synthetic transfer datasets $\{\hat{\mathcal{D}}_k\} = Dec(\{\tilde{\mathcal{E}}_k\})$ from decoder and use them to bridge communication between models. Alongside learning from $\{\hat{\mathcal{D}}_k\}$ in the current round, we also maintain a memory buffer $\boldsymbol{\mathcal{B}}$ to store past synthetic data for later review. The training objectives are defined as follows.

**Knowledge Exchange.** For each client $k$, we minimize the discrepancy between its model's predictions on $\{\hat{\mathcal{D}}_i\}_{i \neq k}$ and the corresponding target logits $\{\hat{Y}_i\}_{i \neq k}$ shared by other clients. The FKE loss is formulated as:

$$\mathcal{L}_{\text{FKE}}^k = \sum_{i=1, i \neq k}^K D_{\text{KL}} \left[ f(\hat{\mathcal{D}}_i; \boldsymbol{\theta}_k) \parallel \hat{Y}_i \right] \tag{13}$$

**Memory Buffer.** Empirically, we observe that knowledge from previous rounds may fade during training (see table 5). To retain the proficiency in past rounds synthetic data, we adopt a memory buffer $\boldsymbol{\mathcal{B}} = \{\mathcal{B}_i\}_{i=1}^K$, where $\mathcal{B}_i$ temporarily stores synthetic data for client $i$ from earlier rounds. Each client's memory buffer is synchronized and contains synthetic data from all clients. In each round $t$, clients sample a subset $\boldsymbol{\mathcal{B}}_t = \{\mathcal{B}_i^t\}_{i=1}^K \subset \boldsymbol{\mathcal{B}}$ to obtain the target logits $\hat{Y}_i^{\mathcal{B}_i^t}$ using their latest model, and share them with others. The knowledge retention loss is defined as:

$$\mathcal{L}_{\text{mem}}^k = \sum_{i=1, i \neq k}^K D_{\text{KL}} \left[ f(\mathcal{B}_i^t; \boldsymbol{\theta}_k) \parallel \hat{Y}_i^{\mathcal{B}_i^t} \right] \tag{14}$$

**Overall Optimization.** In each iteration, clients alternately update their models using the FKE loss on current synthetic data and the knowledge retention loss on memory buffer data. *The pseudocode for Data-Free Federated Knowledge Exchange can be found in the appendix A.*

## 4 EXPERIMENTS

### 4.1 EXPERIMENTAL SETUPS

**Baselines and Model Heterogeneity.** We compare our proposed DFFKE method against nine existing model heterogeneous federated learning algorithms that do not rely on public data. These algorithms, implemented in HtFLlib Zhang et al. (2023), include LG-FedAvg Liang et al. (2020), FedGen Zhu et al. (2021), FedGH Yi et al. (2023), FML Shen et al. (2020), FedKD Wu et al. (2022), FedDistill Jeong et al. (2018), FedProto Tan et al. (2022b), FedTGP Zhang et al. (2024b), and FedKTL Zhang et al. (2024a). Since FedGen originally aggregates client models using FedAvg, HtFLlib implements a modified version combining FedGen with LG-FedAvg, denoted as FedGen†. Note that LG-FedAvg, FedGen†, and FedGH assume a homogeneous classifier. To enable a fair comparison, we only consider model heterogeneity within the feature extractors (encoder) of client models. Following the HtFLlib convention, we use the notation "HtFE-$X$" to represent different heterogeneous model scenarios, where $X$ indicates the number of distinct architectures used among clients. For example, HtFE-1 uses only ResNet18 He et al. (2016), while HtFE-10 includes ten architectures: ResNet18, ResNet34, ResNet50 He et al. (2016), GoogLeNet Szegedy et al. (2015), EfficientNetV2 Tan & Le (2021), MobileNet-v3-small, MobileNet-v3-large Howard et al. (2019), ShuffleNet-v2-x1.5, ShuffleNet-v2-x2.0 Ma et al. (2018), and ViT-Tiny Dosovitskiy et al. (2020). The embedding dimensions $h$ differ across these encoder architectures.

| | Classic Heterogeneous FL | | | | | | | | |
|---|---|---|---|---|---|---|---|---|---|
| | High Data Heterogeneity ($\alpha = 0.1$) | | | Low Data Heterogeneity ($\alpha = 1.0$) | | | | | |
| | TinyImageNet | CIFAR10 | CIFAR100 | TinyImageNet | CIFAR10 | CIFAR100 | CIFAR100 - More Model Heterogeneity | | |
| Methods | HtFE-1 | | | HtFE-1 | | | HtFE-2 | HtFE-5 | HtFE-10 |
| LG-FedAvg | 12.55±0.85 | 33.65±4.25 | 19.57±1.52 | 17.91±0.59 | 63.53±7.29 | 32.34±0.93 | 30.81±0.91 | 29.59±1.91 | 26.35±3.50 |
| FedGen† | 11.98±0.96 | 33.71±4.09 | 19.11±1.48 | 16.99±0.77 | 63.55±6.41 | 31.35±0.87 | 29.27±0.91 | 28.72±1.99 | 25.50±3.13 |
| FedGH | 12.85±1.03 | 34.21±4.39 | 20.05±1.83 | 19.28±0.37 | 64.44±7.64 | 34.09±1.08 | 31.48±1.55 | 30.83±2.18 | 27.74±4.23 |
| FML | 11.81±0.95 | 32.84±4.61 | 18.87±1.51 | 17.66±0.48 | 64.36±7.15 | 32.29±0.85 | 30.40±1.26 | 29.31±1.87 | 26.02±3.31 |
| FedKD | 12.09±1.00 | 32.81±4.61 | 18.76±1.60 | 18.57±0.60 | 63.53±7.98 | 32.35±0.88 | 30.92±1.19 | 29.42±1.89 | 26.73±4.42 |
| FedDistill | 11.84±1.13 | 33.37±4.72 | 18.72±1.44 | 17.50±0.48 | 63.49±8.19 | 31.96±0.94 | 29.55±1.50 | 28.55±2.06 | 25.47±3.41 |
| FedProto | 11.52±0.96 | 34.23±4.57 | 19.68±1.79 | 18.96±0.45 | 63.77±6.46 | 35.93±1.15 | 31.33±1.50 | 30.44±2.49 | 27.77±4.89 |
| FedTGP | 12.48±1.05 | 33.35±4.46 | 19.56±1.69 | 18.75±0.80 | 63.00±7.81 | 33.23±1.13 | 32.24±2.95 | 30.79±3.43 | 28.35±6.93 |
| FedKTL | 10.17±1.01 | 29.19±5.80 | 13.38±1.68 | 14.45±0.61 | 57.83±7.75 | 21.51±2.49 | 18.48±2.58 | 17.40±1.60 | 14.40±7.40 |
| **DFFKE** | **27.92±0.33** | **43.21±4.82** | **38.19±0.76** | **31.74±0.40** | **68.20±4.05** | **47.49±0.38** | **46.48±1.46** | **45.84±1.02** | **39.06±3.43** |

Table 2: Test accuracy (%) of $K = 10$ clients with participation rate $\rho = 1.0$ under different levels of data heterogeneity and heterogeneous model scenarios. Results are reported as the mean and standard deviation of the accuracy of all client models on the **global test set**.

| | Personalized Heterogeneous FL | | | | | | | | |
|---|---|---|---|---|---|---|---|---|---|
| | High Data Heterogeneity ($\alpha = 0.1$) | | | Low Data Heterogeneity ($\alpha = 1.0$) | | | | | |
| | TinyImageNet | CIFAR10 | CIFAR100 | TinyImageNet | CIFAR10 | CIFAR100 | CIFAR100 - Large Client Amount | | |
| Methods | $K = 10, \rho = 1.0$ | | | $K = 10, \rho = 1.0$ | | | $K = 20$ $\rho = 0.5$ | $K = 50$ $\rho = 0.2$ | $K = 100$ $\rho = 0.1$ |
| LG-FedAvg | 55.93±4.18 | 93.49±2.84 | 73.59±4.61 | 32.43±2.44 | 82.70±1.61 | 48.55±3.66 | 39.70±2.65 | 31.37±4.79 | 30.90±5.66 |
| FedGen† | 55.19±4.63 | 93.28±2.74 | 73.19±4.97 | 31.70±2.39 | 82.07±2.46 | 47.72±4.07 | 37.98±2.69 | 31.32±3.97 | 30.55±6.28 |
| FedGH | 57.12±4.58 | 94.11±2.36 | 73.62±5.34 | 32.94±2.32 | 82.05±2.18 | 49.40±3.25 | 39.66±3.53 | 32.35±4.35 | 31.98±5.08 |
| FML | 57.21±4.69 | 94.59±2.62 | 74.04±4.06 | 32.70±2.71 | 83.46±2.06 | 49.89±3.28 | 40.37±2.82 | 32.54±4.34 | 32.33±5.89 |
| FedKD | 56.35±4.72 | 94.55±2.76 | 74.08±4.57 | 32.17±2.33 | 82.87±2.59 | 49.11±4.16 | 39.73±2.62 | 31.85±4.86 | 31.22±5.57 |
| FedDistill | 55.54±4.73 | 94.51±2.52 | 73.86±4.78 | 31.94±2.58 | 82.12±2.23 | 48.34±3.30 | 38.97±2.89 | 30.95±4.27 | 30.77±5.80 |
| FedProto | 54.19±4.00 | 94.24±2.51 | 70.67±4.96 | 32.91±2.18 | 82.84±2.51 | 47.86±3.28 | 38.15±2.59 | 31.25±4.59 | 30.23±5.41 |
| FedTGP | 58.63±3.86 | 94.77±2.50 | 76.52±4.39 | 35.25±2.69 | 83.97±2.01 | 55.22±2.78 | 44.71±3.20 | 35.60±4.61 | 35.40±5.15 |
| FedKTL | 56.84±5.79 | 94.59±3.32 | 74.98±5.53 | 33.20±2.82 | 83.06±3.12 | 51.31±4.58 | 40.15±3.70 | 33.32±4.93 | 33.00±6.26 |
| **DFFKE** | **60.45±3.08** | **95.84±2.37** | **76.95±3.72** | **39.72±1.90** | **84.21±1.63** | **56.83±2.18** | **49.50±2.50** | **43.78±3.91** | **43.63±5.36** |

Table 3: Test accuracy (%) of personalized FL on different numbers of clients and participation rates using HtFE-5. Results are reported as the mean and standard deviation of the accuracy of all client models on their **individual private test sets**.

**Datasets and Data Heterogeneity.** We conduct experiments on CIFAR10/100 Krizhevsky et al. (2009), and TinyImageNet Le & Yang (2015) with heterogeneous data partitions to simulate federated collaborative learning. Following standard practice Lin et al. (2020); Zhu et al. (2021), we use a Dirichlet distribution **Dir**$(\alpha)$ for data partitioning to simulate non-IID distributions among clients. Smaller $\alpha$ values indicate greater data heterogeneity, where each client's private data is biased toward fewer classes from the original dataset. We adopt $\alpha \in \{0.1, 1.0\}$ to represent high and low data heterogeneity, respectively. The training images from all datasets are partitioned using the non-IID method to form clients' private training datasets. For testing, we consider two federated learning (FL) settings: **classic FL** and **personalized FL** Tan et al. (2022a). In classic FL, all clients share an IID test set of 10,000 images to evaluate collaborative learning performance on global knowledge, whereas in personalized FL, each client receives a unique test set that matches the distribution of its private training data, thereby highlighting the benefits of FL for their personal objectives.

**General Implementation Details.** We combine the aforementioned model and data heterogeneity settings to simulate heterogeneous federated learning scenarios. Performance is evaluated by averaging the test accuracy of all clients' models after each round. For all algorithms, we report the highest test accuracy achieved over a maximum of $n = 300$ communication rounds. During training and testing, all image data are resized to $128 \times 128$. We simulate heterogeneous FL scenarios on $K = 10$ clients with a client participation ratio $\rho = 1.0$, and we experiment on $20, 50$, and $100$ clients with $\rho = 0.5, 0.2$, and $0.1$ respectively. For training both the embedding encoder and clients' models in all baseline methods and DFFKE, unless otherwise specified, we use the Adam optimizer with a learning rate of 0.001 and a batch size of 100. To accommodate the consistent embedding dimension assumption in FedGH, FedKD, FedProto, and FedTGP, we follow Zhang et al. (2023) to add an average pooling layer before classifiers and set $h = 512$ by default for all baseline methods.

**DFFKE Implementation Details.** In the DFFKE framework, each client trains locally on its private dataset until converge before communication. During each communication round, the docking

layers are trained for $T_{\text{Tran}} = 100$ epochs. The embedding decoder is trained for $T_{\text{Dec}} = 3, 6$, and 6 epochs on CIFAR10, CIFAR100, and TinyImageNet, respectively, and clients perform knowledge exchange for $T_{\text{FKE}} = 2, 4$, and 4 epochs. For the embedding decoder, we adopt a lightweight 3-layer, 4.4M-parameter architecture from Fang et al. (2021). To evaluate the performance of vanilla DFFKE, differential privacy is disabled by default in main experiments tables.

## 4.2 RESULT COMPARISON

table 2 presents the classic federated learning test accuracy of all methods, where DFFKE consistently outperforms the heterogeneous federated learning baselines by a large margin across all scenarios. Notably, the performance advantage of DFFKE is more pronounced with increasing data heterogeneity and more challenging datasets. For instance, on CIFAR-100 with high data heterogeneity, DFFKE outperforms the best baseline by a substantial margin of **18.14%**. Furthermore, DFFKE also demonstrates competitive performance in the personalized FL task, as shown in table 3. DFFKE outperforms FL baseline methods specialized in the personalized FL setting, such as FedTGP Zhang et al. (2024b). Across varying client counts $K$ and participation rates $\rho$, DFFKE consistently maintains higher performance and stability than the baseline methods.

## 4.3 DFFKE ANALYSIS

In this section, we evaluate the effectiveness of the design components introduced in DFFKE and computation overhead. Unless otherwise specified, all experiments are conducted with $K = 10$ clients, a participation rate of $\rho = 1.0$, on the CIFAR100 dataset with a data heterogeneity parameter of $\alpha = 1.0$, and using the HtFE-1 model group. *More ablation studies can be found in the Appendix.*

**Differential Privacy.** We propose to leverage clients' local embeddings and logits to improve the effectiveness of DFFKE, but this could expose clients to the risk of data leakage through malicious reverse engineering attack. To address this issue, we suggest applying additive Gaussian noise based on $(\epsilon, \delta)$-Differential Privacy to obscure private information. Specifically, Differential Privacy (DP) Dwork et al. (2014) is a theoretically proven framework for releasing statistical information about datasets while protecting the privacy of individual data samples. It has been widely adopted in deep learning Abadi et al. (2016); Zhao et al. (2019) and federated learning Wei et al. (2020); El Ouadrhiri & Abdelhadi (2022) tasks to protect individual data while preserving the utility of the released data. We propose to obscure private information in the embeddings or logits by adding Gaussian noise following the $(\epsilon, \delta)$-Differential Privacy standard, as given by:

$$\boldsymbol{x}_{encrypted} = \boldsymbol{x} + \mathcal{N}\left(0, \frac{2\ln(1.25/\delta)(\Delta f)^2}{\epsilon^2} I_{|\boldsymbol{x}|}\right) \tag{15}$$

where the privacy budget $\epsilon \leq 1$ controls the trade-off between privacy and utility, $\delta \leq 1$ represents the failure probability of the differential privacy guarantee, and $\Delta f$ represents the model sensitivity. For DFFKE, we set $\epsilon = 1$ and $\delta = \frac{1}{\text{Dataset Size}}$ to ensure that every data point is protected. In addition, we present an sensitivity analysis w.r.t. noise level in table 4. DP effectively reduces privacy risk,

| Dataset | $\alpha$ | Best Baseline | w/o DP $C_{\mathcal{E}} = 1$ $C_{\hat{Y}} = 1$ | w/ DP Base $\epsilon = 1$ $C_{\mathcal{E}} = 0.45$ $C_{\hat{Y}} = 0.67$ | w/ DP+ $\epsilon = 0.75$ $C_{\mathcal{E}} = 0.35$ $C_{\hat{Y}} = 0.44$ | w/ DP++ $\epsilon = 0.5$ $C_{\mathcal{E}} = 0.24$ $C_{\hat{Y}} = 0.27$ | w/ DP+++ $\epsilon = 0.25$ $C_{\mathcal{E}} = 0.12$ $C_{\hat{Y}} = 0.13$ | Pure Noise $C_{\mathcal{E}} = 0$ $C_{\hat{Y}} = 0$ |
|---|---|---|---|---|---|---|---|---|
| CIFAR100 | 0.1 | 20.05±1.83 | 38.19±0.76 | 36.67±1.03 | 35.22±0.63 | 34.85±0.92 | 34.92±0.72 | 34.58±1.12 |
| | 1.0 | 35.93±1.15 | 47.49±0.38 | 45.60±0.57 | 44.98±0.55 | 45.03±0.89 | 44.81±0.66 | 44.88±0.32 |
| CIFAR10 | 0.1 | 34.23±4.57 | 43.21±4.82 | 42.24±6.42 | 41.92±5.77 | 41.68±6.03 | 41.52±5.14 | 41.71±5.59 |
| | 1.0 | 64.44±7.64 | 68.20±4.05 | 68.15±3.43 | 67.48±2.95 | 66.44±3.27 | 66.04±3.94 | 66.35±3.71 |
| TinyImageNet | 0.1 | 12.85±1.03 | 27.92±0.33 | 26.36±0.55 | 25.85±0.43 | 25.39±0.51 | 25.55±0.37 | 25.40±0.59 |
| | 1.0 | 19.28±0.37 | 31.74±0.40 | 32.17±0.25 | 31.73±0.49 | 31.77±0.43 | 31.92±0.53 | 31.59±0.39 |

Table 4: Sensitivity analysis w.r.t. different noise level in differential privacy, conducted by varying the privacy budget $\epsilon \in \{1.0, 0.75, 0.5, 0.25\}$ or by replacing the embedding $\mathcal{E}$ and logit $\hat{Y}$ entirely with noise. Experiments are conducted in the Classic FL setting. To interpret the effect of injected noise, we report $C_{\mathcal{E}}$ and $C_{\hat{Y}}$, which denote the cosine similarities between $\mathcal{E}$ and $\mathcal{E}_{encrypted}$, and between $\hat{Y}$ and $\hat{Y}_{encrypted}$, respectively.

and **the analysis result shows a clear privacy–utility trade-off**. When pushing noise to extreme level, which fundamentally replacing embeddings and logits by pure noise, DFFKE still maintains significant superiority over the baseline methods. As such, **sharing clients' private embeddings and logits are not mandatory in our framework.** In real-world FL scenario, users can opt in sharing such information based on individual privacy preference with protection of DP. *For the proof of $(\epsilon, \delta)$-Differential Privacy using the Gaussian Mechanism and additional details about the model sensitivity $\Delta f$, please refer to the appendix G.*

**Effectiveness of Memory Buffer.** As shown in table 5, the memory Buffer is a fundamental component in DFFKE. We observe that removing the memory Buffer leads to a performance degradation of $9.27\%$. Additionally, we evaluate DFFKE's performance by varying the memory limit from 1 round up to no limit (i.e., storing all past synthetic data). The results indicate that test accuracy correlates with the size of the memory Buffer, demonstrating that retaining previous round synthetic data is essential to maximize DFFKE performance.

| Memory Buffer Size | Accuracy(%) |
|---|---|
| No Memory Buffer | $38.22 \pm 0.90$ |
| Memory Limit 1 | $41.59 \pm 0.77$ |
| Memory Limit 5 | $44.18 \pm 0.60$ |
| Memory Limit 10 | $45.01 \pm 0.96$ |
| Memory Limit 20 | $46.67 \pm 0.43$ |
| DFFKE (Unlimited, up to 50) | $47.49 \pm 0.38$ |

Table 5: Impact of memory buffer size. (the number of past communication rounds' synthetic data stored.)

**Computation and Communication Cost.** DFFKE achieves superior performance without compromising computational efficiency as shown in fig. 3. In addition, DFFKE consumes a total of 47.34 GB communication overhead, which remains at the same level of upload/download overhead as the widely used FL approaches. For example, FedAvg McMahan et al. (2017) uploads/downloads model weights each round and converges in 200 rounds of communication and uses a total of 171.4 GB of traffic under the same experiment setting. Also, comparing with other Knowledge Distillation (KD) based FL approaches such as FedKD and FML, they incur much higher total costs than our method (251.32 GB and 559.18 GB, respectively). For the majority of federated learning methods, which transfer client models $\boldsymbol{\theta}$ with a time complexity of $O(m)$, where $m$ is the model size $|\boldsymbol{\theta}|$, DFFKE maintains the same $O(m)$ complexity by uploading local models $\boldsymbol{\theta}_k$ and downloading a lightweight decoder $Dec$ with $|Dec| < m$. *See the appendix D for a more detailed communication cost analysis.*

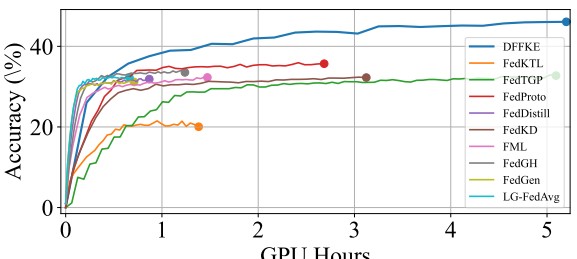

Figure 3: Computation cost comparison. Each algorithm is running on a single RTX 4090.

**Theoretical Analysis.** The theoretical analysis of DFFKE in addressing data heterogeneity can be found in appendix H.

## 5 CONCLUSION

In this paper, we propose Federated Knowledge Exchange (FKE), a novel aggregation-free learning paradigm for heterogeneous federated learning (HFL). By applying FKE to address data and model heterogeneity, we eliminate the need for a global model and enable direct knowledge sharing between clients. Compared to traditional two-step knowledge distillation approaches in FL, which require an intermediate global model to aggregate knowledge and redistribute, direct knowledge exchange preserves more accurate information and reduces potential information loss. To remove reliance on public data for knowledge transfer, we attach a lightweight embedding decoder that produces transfer data, forming the Data-Free Federated Knowledge Exchange (DFFKE) framework. Extensive experiments demonstrate that DFFKE achieves superior performance without compromising computational efficiency, communication cost, or client privacy.

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

# APPENDIX

## A  DFFKE PSEUDOCODE

---

**Algorithm 1** Data-Free Federated Knowledge Exchange

---

1: **Require**: Clients' private datasets $\{\mathcal{D}_k\}_{k=1}^K$, Heterogeneous models $\{\boldsymbol{\theta}_k\}_{k=1}^K$, Generator model $Gen$, Generator training steps $T_{\text{Gen}}$, FKE steps $T_{\text{FKE}}$.
2: **for** each communication round $t$ **do**
3:     *# Local Training:*
4:     **for** each client $k \in K$ **in parallel do**
5:         Local training $\boldsymbol{\theta}_k$ on private data $\mathcal{D}_k$ by Eq. 9
6:         Share $\boldsymbol{\theta}_k, \mathcal{E}_k$ to Data-Free Module
7:         Apply $(\epsilon, \delta)$-Differential Privacy on $\mathcal{E}_k$ *(Optional)*

8:     *# Embedding Space Unification:*
9:     Optimize docking layers $\{\boldsymbol{\theta}_k^z\}_{k=1}^K$ using Eq. 11
10:    Translate embeddings $\tilde{\mathcal{E}}_k = z(\mathcal{E}_k; \boldsymbol{\theta}_k^z)$ for all clients
11:    *# Synthetic Representation Space:*
12:    Train $Gen$ by minimizing Eq. 12 for $T_{\text{Gen}}$ steps
13:    Distribute $Dec$ and $\{\tilde{\mathcal{E}}_k\}_{k=1}^K$ to all clients
14:    *# Federated Knowledge Exchange:*
15:    **for** each client $k \in K$ **in parallel do**
16:        Collect synthetic data $\hat{\mathcal{D}}_i = Dec(\tilde{\mathcal{E}}_i)$ for all $i \neq k$
17:        Sample a subset $\boldsymbol{\mathcal{B}}_t$ and share logits output $\hat{Y}_k^{\mathcal{B}_k^t}$
18:        **for** $s = 1, \dots, T_{\text{FKE}}$ **do**
19:            Knowledge exchange by minimizing Eq. 13
20:            Review memory buffer by minimizing Eq. 14
21:    Update memory bank $\mathcal{B}_k$ with new synthetic data

---

## B  TABLE OF NOTATIONS

| Notations | Definitions or Descriptions |
|---|---|
| $K$ | the number of participating clients in a round |
| $c_k$ | the $k$-th client |
| $\mathcal{D}_k$ | the private dataset belongs to $c_k$ |
| $(X_k, Y_k)$ | the data samples and labels in $\mathcal{D}_k$ |
| $N_k$ | the size of private dataset $\mathcal{D}_k$ |
| $\hat{\mathcal{D}}_k$ | the transfer dataset assigned to $c_k$ |
| $f(\cdot), \boldsymbol{\theta}_k$ | full network and $c_k$'s model parameter |
| $h(\cdot), \boldsymbol{\theta}_k^h$ | encoder and $c_k$'s encoder parameter |
| $l(\cdot), \boldsymbol{\theta}_k^l$ | classifier and $c_k$'s classifier parameter |
| $z(\cdot), \boldsymbol{\theta}_k^z$ | translator and $c_k$'s translator parameter |
| $\mathcal{E}_k$ | embedding collection of $c_k$'s data |
| $\hat{Y}_k$ | logit collection of $c_k$'s data |
| $\tilde{\mathcal{E}}_k$ | translated embeddings from $\mathcal{E}_k$ |
| $\{\boldsymbol{\theta}_k\}$ | set of all clients' model parameters |
| $\{\mathcal{E}_k\}$ | set of all clients' embeddings |
| $\{\hat{Y}_k\}$ | set of all clients' logits |
| $\boldsymbol{\mathcal{B}}^t$ | subset of memory buffer for round $t$ |

Table 6: Notations used in this paper.

## C   ABLATION: LIMITED CLIENT PARTICIPATION

| DFFKE Participation Proportion $\pi$ | Accuracy(%) |
|---|---|
| 10%, No Collaboration (1 out of 10) | 31.25±0.85 |
| 20% (2 out of 10) | 35.71±0.20 |
| 30% (3 out of 10) | 39.94±0.08 |
| 40% (4 out of 10) | 42.43±0.30 |
| 50% (5 out of 10) | 43.98±0.51 |
| 60% (6 out of 10) | 45.79±0.45 |
| 70% (7 out of 10) | 46.57±0.41 |
| 80% (8 out of 10) | 47.00±0.32 |
| 90% (9 out of 10) | 47.32±0.60 |
| 100%, Full Participation (10 out of 10) | 47.49±0.38 |

Table 7: CIFAR100 is partitioned into 10 shares with low data heterogeneity $\alpha = 1.0$. Each client owning $1/10$ of the total data.

Our main experiments are conducted under the hypothesis of perfect collaboration, where all clients participate in the FL at least once. To evaluate DFFKE's performance under limited collaboration, we conduct an ablation study on the participation proportion $\pi$, as shown in table 7. The average test accuracy grows as $\pi$ increase from $10\%$ to $100\%$. This aligns with the intuition that larger collaboration scales benefit clients by providing access to a broader global knowledge base. Notably, the improvement slows and plateaus after $\rho = 70\%$, suggesting that benefit of scaling DFFKE is diminishing.

## D   ADDITIONAL COMMUNICATION COST ANALYSIS

As discussed in section 4.3: Communication Cost, we state that (1) DFFKE maintain a same space complexity of $O(m)$ for transferring models, where $m$ is the model size $|\boldsymbol{\theta}|$. (2) Existing KD approaches transfer knowledge through embeddings $\mathcal{E}$ and logits $\hat{Y}$, with a time complexity of $O(\hat{n})$, where $\hat{n}$ is the transfer dataset size $|\hat{\mathcal{D}}|$. In comparison, DFFKE performs two rounds of communication involving the sets $\{\mathcal{E}_k\}_{k=1}^K$ or $\{\hat{Y}_k\}_{k=1}^K$, resulting in the same overall complexity of $O(\hat{n})$, where $K$ is the number of clients and $|\{\hat{\mathcal{D}}_k\}_{k=1}^K| = |\hat{\mathcal{D}}|$. This holds particularly when the number of clients participating in each round remains fixed, even as the total number of clients increases. Specifically, DFFKE involves two circulations of embeddings and logits per round: (1) When training Data-Free Module, each client $k$ uploads $\mathcal{E}_k$ and receives $\{\tilde{\mathcal{E}}_k\}_{k=1}^K$ in return. (2) During knowledge exchange, each client $k$ share $\hat{Y}_k^{\hat{\mathcal{D}}_k}, \hat{Y}_k^{\mathcal{B}_k^t}$ to $K-1$ other clients and receives a set $\{\hat{Y}_i^{\hat{\mathcal{D}}_i}, \hat{Y}_i^{\mathcal{B}_i^t}\}_{i=1, i\neq k}^K$ from all other clients, where $|\{\hat{\mathcal{B}}_k^t\}_{k=1}^K| = \hat{n}$. Therefore, since each transmission is bounded by $O(\hat{n})$, the overall complexity for each client remains $O(\hat{n})$.

## E   NECESSITY OF THE MODEL DISCREPANCY LOSS FOR EMBEDDING DECODER.

Training the generative model in combination with a model discrepancy loss $\mathcal{L}_{md}$ (eq. (4)) to produce a synthetic dataset is a common approach in data-free knowledge distillation. However, in the data-free module of DFFKE, we found that $\mathcal{L}_{md}$ is not beneficial. table 8 presents an ablation study of various function choices $F$ for $\mathcal{L}_{md}$ during our embedding decoder training. Notably, when $F$ is implemented using KL-Divergence (*KL-Div*), the loss value quickly diverges to $-\infty$, preventing algorithm convergence. To address this, we scale the loss by a coefficient of 0.1 to moderate its effect. *MAE* and *MSE* are commonly used as alternatives for the model discrepancy loss in previous approaches Zhang et al. (2022b), as their values are bounded within $[-\frac{2}{n}, 0]$, where $n$ is the number of classes. Our results indicate that incorporating the model discrepancy loss does not significantly improve the performance of DFFKE.

| $F$ Choice for Model Discrepancy Loss | Accuracy (%) |
|---|---|
| No Model Discrepancy Loss | 47.49±0.38 |
| KL-Divergence (*KL-Div*) | 46.54±0.55 |
| Mean Absolute Difference (*MAE*) | 47.08±0.64 |
| Mean Squared Difference (*MSE*) | 47.56±0.48 |

Table 8: Test accuracy (%) of DFFKE in the classic FL setting with different design choices for $\mathcal{L}_{\text{md}}$ in embedding decoder training.

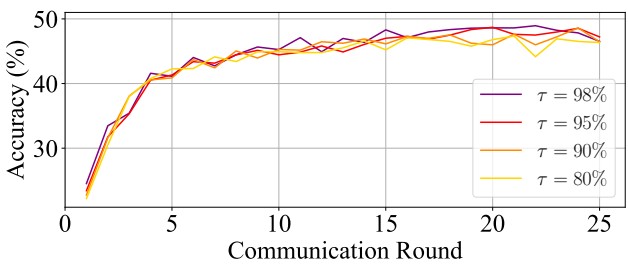

ht

Figure 4: DFFKE is robust against variations in local training epochs (Varied by local training goal $\tau$).

## F    IMPACT OF PRIVATE TRAINING EPOCHS.

**Impact of Local Training Epochs.**    In DFFKE, we set a training accuracy goal $\tau$ for clients during private training instead of specifying a fixed number of epochs. The idea is that overfitting individual models to their private datasets strengthens their expertise, thereby enhancing the effectiveness of knowledge exchange. The local training accuracy goal $\tau$ can range from $0.8$ to $0.98$, and DFFKE is highly tolerant to variations in $\tau$ (see fig. 4 and table 9). The difference in test accuracy between goals $\tau = 98\%$ and $\tau = 80\%$ is only 1.6%, with nearly identical convergence speeds.

| Goal | $E_{\text{local-training}}$ | Accuracy(%) |
|---|---|---|
| $\tau = 80\%$ | (30, 7, 4, 3, 3) | 45.89±0.58 |
| $\tau = 90\%$ | (40, 9, 5, 5, 4) | 46.85±0.56 |
| $\tau = 95\%$ | (60, 12, 7, 6, 5) | 47.25±0.49 |
| $\tau = 98\%$ | (80, 14, 10, 8, 7) | 47.49±0.38 |

Table 9: Impact of different numbers of private training epochs $E$ (approximated empirically based on accuracy goal $\tau$). $E_{\text{local-training}}$ is presented as a list of local training epochs sampled from communication rounds $t = (0, 1, 5, 10, 20)$ respectively.

## G    DIFFERENTIAL PRIVACY PROOF FOR GAUSSIAN MECHANISM

To prove a gaussian mechanism $\mathcal{M}$ is $(\epsilon, \delta)$-differentially private, we need to show that for any measurable set $S \subseteq \mathbb{R}^k$ and for all neighboring data sample $X_1$ and $X_2$, $\mathcal{M}$ satisfies

$$\Pr[\mathcal{M}(X_1) \in S] \leq e^\epsilon \Pr[\mathcal{M}(X_2) \in S] + \delta.$$

***Proof.***    Let $f : \mathcal{X} \to \mathbb{R}^k$ be a vision model with sensitivity,

$$\|f(X_1) - f(X_2)\|_2 \leq \Delta f,$$

for all neighboring data points $X_1$ and $X_2$. The Gaussian mechanism is defined by

$$\mathcal{M}(X) = f(X) + \mathcal{N}\left(0, \frac{2\ln(1.25/\delta)(\Delta f)^2}{\epsilon^2} I_k\right),$$

The probability density function of $\mathcal{M}(X_1)$ is given by

$$p(\mathcal{E}) = \frac{1}{(2\pi\sigma^2)^{k/2}} \exp\left(-\frac{\|\mathcal{E} - f(X_1)\|_2^2}{2\sigma^2}\right),$$

and for $\mathcal{M}(X_2)$,

$$p'(\mathcal{E}) = \frac{1}{(2\pi\sigma^2)^{k/2}} \exp\left(-\frac{\|\mathcal{E} - f(X_2)\|_2^2}{2\sigma^2}\right).$$

*Privacy Loss:* Define the privacy loss at output $\mathcal{E}$ as

$$L(\mathcal{E}) = \ln\frac{p(\mathcal{E})}{p'(\mathcal{E})} = \frac{\|\mathcal{E} - f(X_2)\|_2^2 - \|\mathcal{E} - f(X_1)\|_2^2}{2\sigma^2}.$$

*Bounding the Privacy Loss:* Using the bound on the sensitivity, it can be shown Dwork et al. (2014) that the tail probability of the privacy loss satisfies

$$\Pr\left[L(\mathcal{E}) > \epsilon\right] \leq \delta.$$

This is achieved by analyzing the difference $\|\mathcal{E} - f(X_2)\|_2^2 - \|\mathcal{E} - f(X_1)\|_2^2$ and using properties of the Gaussian distribution.

With the chosen $\sigma$, the mechanism $\mathcal{M}$ satisfies

$$\Pr[\mathcal{M}(X_1) \in S] \leq e^\epsilon \Pr[\mathcal{M}(X_2) \in S] + \delta.$$

Thus, the Gaussian mechanism with

$$\sigma^2 = \frac{2\ln(1.25/\delta)(\Delta f)^2}{\epsilon^2} I_k$$

satisfies $(\epsilon, \delta)$-Differential Privacy.

**Model Sensitivity.** The sensitivity $\Delta f$ measures the maximum change in a function's output when a single data point is modified or removed. For image data, such a change is typically manifested as a modification to a pixel value. In practice, we estimate the model sensitivity by selecting a random pixel in an image and replacing its value with a random value sampled from the uniform distribution $U(0, 255)$.

## H  THEORETICAL ANALYSIS OF DFFKE IN ADDRESSING DATA HETEROGENEITY

This section shows that **DFFKE** reduces the adverse effect of Non-IID data by (i) learning a *synthetic distribution* that matches the global mixture and (ii) exchanging knowledge so that every client asymptotically minimises its risk on that mixture. Throughout, let

$$p_k(x) \quad \text{be the data distribution of client } k, \qquad p(x) = \frac{1}{K}\sum_{k=1}^{K} p_k(x)$$

and let $p_{\text{syn}}(x)$ denote the distribution induced by the embedding decoder after training.

### H.1  ASSUMPTIONS

1. **Embedding alignment.** For every class $y$, the docking layers $\{z_k\}$ satisfy $z_k\big(h(x; \boldsymbol{\theta}_k^h)\big) \sim q_y$ for all $k$, where $q_y$ is the embedding distribution of class $y$ shared amount all clients in the unified embedding space. Alignment loss equation 11 enforces this.

2. **Decoder fidelity.** The decoder $Dec$ is trained to minimize $\mathcal{L}_{\text{fid}}$ in equation 12. Hence $Dec$ is the left inverse of the aligned encoder on the support of every $q_y$.

3. **Finite capacity and uniform mixing.** Each client uses at most $N_k$ synthetic samples per round and shares them with all other clients through the memory buffer, so that every client observes $N_k(K-1)$ IID draws from $p_{\text{syn}}$ per communication round.

## H.2 Main Results

**Theorem 1** (Synthetic distribution consistency). *Under Assumptions 1–2, the optimal decoder yields*

$$p_{\text{syn}}(x) = p(x).$$

*Proof.* Fix a class $y$. Let $\varepsilon \sim q_y$ be an aligned embedding obtained from any client. Because the decoder is a left inverse, $Dec(\varepsilon)$ is a sample whose re-encoded embedding again follows $q_y$. Hence the joint distribution of $(x, y)$ generated by $(Dec, q_y)$ equals the union of all client distributions conditioned on $y$. Marginalising over $y$, and since each client contributes equally through the uniform mixing protocol, we obtain $p_{\text{syn}}(x) = \frac{1}{K} \sum_k p_k(x) = p(x)$. $\square$

**Theorem 2** (Generalisation bound of DFFKE). *Let $f_k^{(T)}$ be the model of client $k$ after $T$ communication rounds, and let*

$$L(f) = \mathbb{E}_{x \sim p}[\ell(f(x), y)]$$

*be the expected cross-entropy loss on the global mixture. Suppose that each client follows the FKE training objective equation 13 with learning rate $\eta$ and that $\ell$ is 1-Lipschitz and bounded in $[0, 1]$. Then, with probability at least $1 - \delta$,*

$$\frac{1}{K} \sum_{k=1}^{K} L(f_k^{(T)}) \leq \frac{1}{K} \sum_{k=1}^{K} \widehat{L}_k^{\text{syn}} + \sqrt{\frac{\log(2/\delta)}{2N_k(K-1)T}} + \eta \underbrace{\left(\varepsilon_{\text{dec}} + \varepsilon_{\text{mem}}\right)}_{\text{bias terms}},$$

*where $\widehat{L}_k^{\text{syn}}$ is the empirical loss of $f_k^{(T)}$ on its synthetic mini-batches, $\varepsilon_{\text{dec}} = \sup_x \|x - Dec(z_k(h(x)))\|$ measures residual decoder error, and $\varepsilon_{\text{mem}}$ measures imperfect coverage of past rounds.*

*Sketch proof.* Because Theorem 1 grants $p_{\text{syn}} = p$, each synthetic mini-batch is an IID sample from the target distribution. A standard uniform convergence argument (generalization bound by Hoeffding inequality Hoeffding (1963)) yields the concentration term $\sqrt{\log(2/\delta)/(2N_k(K-1)T)}$. Optimization dynamics under stochastic gradient descent with learning rate $\eta$ add a bias that scales with the magnitude of the residual decoder error and the memory buffer mismatch, completing the bound. $\square$

**Discussion.** Theorems 1–2 show that, once the decoder fidelity is high and the memory buffer is large enough, the additional bias terms vanish. The remaining bound is identical to that of a centrally trained model on $p(x)$, and no term depends on any divergence between $p_k$ and $p$. Hence DFFKE removes the Non-IID penalty that appears in prior bounds for KD-based or model-aggregation methods.

**Corollary.** *When $\varepsilon_{\text{dec}} \to 0$ and $\varepsilon_{\text{mem}} \to 0$, every client's model converges (in expectation) to the minimizer of $L(f)$, matching the optimal centralized solution.*

## I LLM Usage

We applied LLM to check grammar and polish writing.

## J Data Heterogeneity Visualization

fig. 5, 6, and 7 visualize the heterogeneous partitions of CIFAR10 and CIFAR100 used in our experiments. The size of the circle corresponds to the number of data samples.

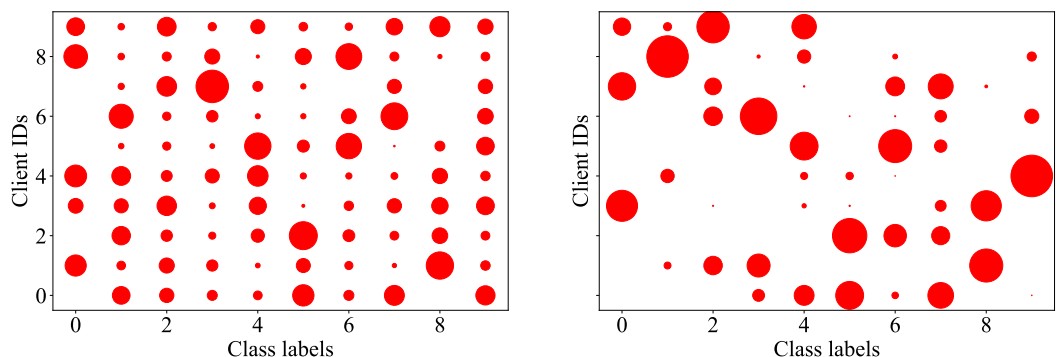

Figure 5: **Left:** 10 Clients, CIFAR10, Low Data-Hetero ($\alpha = 1.0$). **Right:** 10 Clients, CIFAR10, High Data-Hetero ($\alpha = 0.1$).

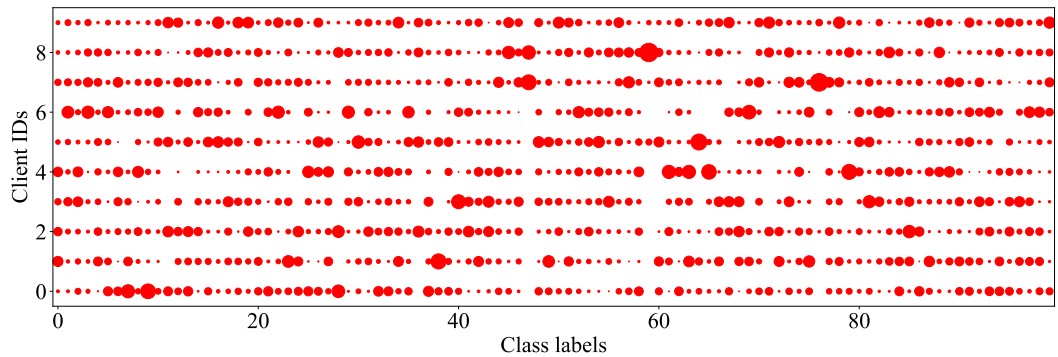

Figure 6: 10 Clients, CIFAR100, Low Data-Hetero ($\alpha = 1.0$)

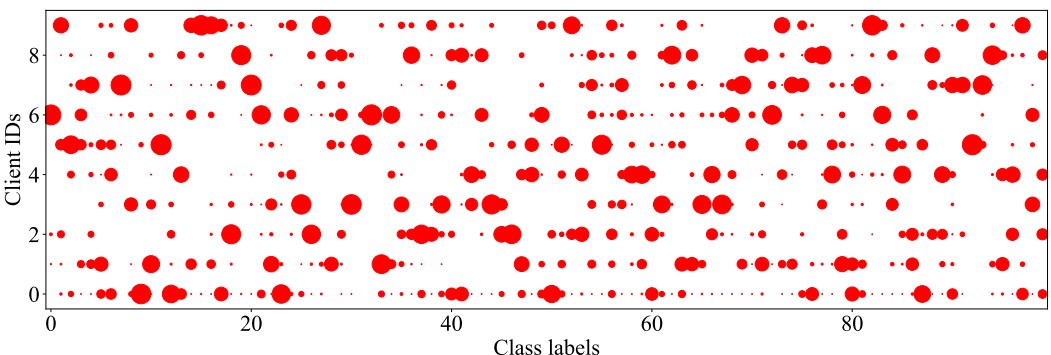

Figure 7: 10 Clients, CIFAR100, High Data-Hetero ($\alpha = 0.1$)