# OpenReview forum: "Data-Free Knowledge Exchange for Aggregation-Free Heterogeneous Federated Learning"
_ICLR.cc/2026/Conference — ICLR 2026 Conference Withdrawn Submission_

### Official Review · Reviewer_EQ9a · 2025-10-28

**Soundness:** 2
**Presentation:** 2
**Contribution:** 1
**Rating:** 4
**Confidence:** 5

**Summary:**

This paper proposes a novel Federated Learning (FL) framework named Data-Free Federated Knowledge Exchange (DFFKE). The core innovation lies in its aggregation-free paradigm, which facilitates direct knowledge exchange between clients without relying on a global model. It tackles both data and model heterogeneity by introducing three key components: an embedding space unification mechanism using a global classifier, a lightweight embedding decoder for generating synthetic data, and a memory buffer to mitigate knowledge forgetting. The framework is designed to operate without public datasets and supports heterogeneous client model architectures. Extensive experiments demonstrate that DFFKE achieves superior performance compared to several state-of-the-art baselines across various FL settings.

**Strengths:**

1. It proposes a novel FL framework that eliminates the need for public datasets, model aggregation, or knowledge aggregation into a global model.

2. It supports model heterogeneity, allowing clients to employ architectures tailored to their local data without any form of model parameter exchange.

**Weaknesses:**

1. The claimed novelty over existing methods is questionable. Data-free knowledge distillation (DFKD) based FL methods that do not rely on public datasets were proposed as early as 2021. Furthermore, methods like FedIOD [1] (2023) already support model heterogeneity and data-free knowledge distillation. In comparison, DFFKE's design, which stacks components like a global classifier, an embedding decoder, and a memory buffer, appears to be a complex amalgamation of existing techniques rather than a fundamentally novel breakthrough.

2. The requirement for clients to share their data embeddings with the central server for space unification and decoder training presents a significant privacy concern. Embeddings are projections of raw data in a high-dimensional feature space and could be leveraged by a malicious server or other clients to infer sensitive attributes of the original data through model inversion or membership inference attacks. While the authors suggest applying differential privacy (DP) or opting out of sharing, the effectiveness of the L_fid loss alone in generating high-quality, knowledge-transferable synthetic data from heavily noised or absent embeddings is doubtful and not sufficiently proven.

3. The framework's performance is highly dependent on the successful alignment of the client embedding spaces. If this unification is suboptimal, the quality of the synthetic data generated by the decoder will degrade, subsequently impairing the knowledge exchange process.

4. The memory buffer, while crucial for performance, introduces additional storage and computational overhead for clients. The buffer size is a critical hyperparameter that requires careful tuning, as a small buffer leads to knowledge forgetting while a large one increases the client's burden. The marginal performance gain on CIFAR-10 in Table 3, despite this added complexity, raises questions about the cost-benefit trade-off in certain scenarios.

5. The experimental evaluation lacks comparison with several prominent and high-performing DFKD-based FL methods, such as FedFTG [2] and DENSE [3]. Their absence makes it difficult to comprehensively assess DFFKE's standing within the most relevant field of research.

[1] Xuan Gong, et al. Federated Learning via Input-Output Collaborative Distillation.

[2] Lin Zhang, et al. Fine-tuning Global Model via Data-Free Knowledge Distillation for Non-IID Federated Learning.

[3] Jie Zhang, et al. DENSE: Data-Free One-Shot Federated Learning.

**Questions:**

1. On page 8, Table 4 reports that on the TinyImageNet dataset with α=1.0, adding DP noise seemingly improves performance compared to the no-DP baseline (from 31.74% to 32.17%). Could you please explain this counter-intuitive result? Does the injected noise act as a regularizer in this specific context, and if so, why is this effect not consistently observed across all datasets and heterogeneity settings?

2. Given that the memory buffer introduces non-trivial client-side burden, how do you recommend practitioners strategically determine the optimal buffer size to balance performance gains against storage and computational costs, especially considering the seemingly modest improvements on datasets like CIFAR-10?

3. What is the rationale for not including comparisons with contemporary and high-performing DFKD-FL methods like FedFTG and DENSE? A comparison with these methods is crucial to firmly establish the state-of-the-art performance of DFFKE.

4. The paper removes the commonly used diversity loss from traditional DFKD, relying primarily on the fidelity loss (L_fid). Without an explicit diversity-promoting loss, how does your method effectively ensure the diversity of the generated synthetic samples, which is critical for comprehensive knowledge transfer?

5. Could you provide visual examples (e.g., in the appendix) of what these synthetic samples look like? This would help in assessing their quality and semantic meaningfulness.

6. Does the act of distributing the generator/embeddings to create a common set of synthetic samples across all clients introduce a new attack vector or privacy concern, even if the raw data is never shared?

---

### Official Review · Reviewer_dgdU · 2025-10-31

**Soundness:** 3
**Presentation:** 4
**Contribution:** 3
**Rating:** 6
**Confidence:** 4

**Summary:**

This paper proposes Federated Knowledge Exchange (FKE), an aggregation-free FL method designed for data and model heterogeneity. Building on the challenge of effective knowledge transfer without access to shared data or requiring model aggregation, the authors propose the Data-Free Federated Knowledge Exchange (DFFKE) framework, where each client both teaches and learns directly from peers via synthetic data generated by a shared embedding decoder. The framework utilizes a unification mechanism for embedding spaces and employs differential privacy to support privacy-preserving knowledge exchange. Extensive experimental results on standard federated learning benchmarks demonstrate that DFFKE outperforms multiple state-of-the-art methods in both classic and personalized heterogeneous FL scenarios.

**Strengths:**

1. DFFKE eliminates the dependency on public data, which is a long-standing challenge in realistic FL scenarios. Furthermore, its aggregation-free design and support for model heterogeneity allow it to adapt effectively to complex FL environments, demonstrating strong real-world applicability.
2. DFFKE is consistently superior to the best available competitors, often by wide margins (e.g., +18.14% over the best baseline on CIFAR-100 under severe heterogeneity in Table 2). In particular, after incorporating the differential privacy mechanism, the performance remains consistently superior to the baselines, indicating that the framework effectively facilitates the exchange of private knowledge among clients and enables joint model optimization.
3. The paper is well-written, with a clear line of reasoning, comprehensive analysis, and solid theoretical guarantees.

**Weaknesses:**

- Only toy-level datasets (CIFAR-10, CIFAR-100, and TinyImageNet) are used. More challenging datasets (such as DomainNet) are needed for the evaluation. Moreover, the study considers only the image modality; incorporating additional modalities would be necessary to demonstrate the generality of the proposed method.
- Although Section 4.3 discusses the integration of differential privacy, the privacy guarantees—particularly against known attacks such as membership inference, gradient inversion, and reconstruction attacks—are neither empirically tested nor demonstrated, leaving the privacy effectiveness partly speculative. Therefore, an empirical evaluation of DP’s protective capability against specific privacy attacks is needed.
- As Figure 2 and Table 5 show, DFFKE’s performance depends heavily on an unbounded or very large memory buffer of synthetic data. This raises concerns regarding storage demands and potential privacy leakage, especially as the number of clients or rounds increases. Memory buffer management strategies are not discussed (e.g., eviction, sampling, privacy-compromised retention).

**Questions:**

- Since the datasets and models used in the experiments are relatively simple, the cost of transmitting embeddings and logits is not significant. However, when the number of classes becomes very large, for example, 100,000, the communication cost of transmitting logits would increase dramatically (as more logits need to be transmitted and each individual logit becomes larger), potentially even exceeding the cost of transmitting the entire model. How does this method address or consider this issue?
- The reviewer may miss some details, but why are experiments with model heterogeneity not conducted under strong data heterogeneity (as in Table 2)? Additionally, why are experiments with a large number of clients not performed under strong data heterogeneity (as in Table 3)?

---

### Official Review · Reviewer_B22i · 2025-11-01

**Soundness:** 2
**Presentation:** 3
**Contribution:** 2
**Rating:** 4
**Confidence:** 4

**Summary:**

The authors propose Federated Knowledge Exchange (FKE) to address data and model heterogeneity in federated learning. FKE consists of four procedures in each communication round: knowledge sharing, embedding space unification, synthetic data generation, and federated knowledge exchange. The authors have conducted extensive experiments to verify the effectiveness of the proposed method.

**Strengths:**

1. The method is clearly presented, and the paper is well-organized and easy to follow.

2. The proposed method is technically sound.

3. Extensive experiments have been conducted under different settings.

**Weaknesses:**

1. The novelty of the proposed method is a major concern, as bidirectional knowledge distillation is a well-established technique. Moreover, several data-free federated knowledge distillation approaches have been proposed in recent years. It is therefore unclear what the key insight and technical novelty of this paper are.

2 Some experimental settings are unclear. For example, how is model heterogeneity tested?

**Questions:**

See weaknesses.

---

### Official Review · Reviewer_dBun · 2025-11-02

**Soundness:** 2
**Presentation:** 3
**Contribution:** 2
**Rating:** 4
**Confidence:** 4

**Summary:**

This paper introduces Data-Free Federated Knowledge Exchange (DFFKE), a novel aggregation-free framework designed for heterogeneous federated learning that eliminates the need for both a global model and public datasets by enabling clients to directly exchange knowledge with each other using synthetic data generated from a lightweight embedding decoder, and extensive experiments demonstrate that DFFKE significantly outperforms nine state-of-the-art baselines by up to 18.14% across various levels of data and model heterogeneity while maintaining computational and communication efficiency.

**Strengths:**

1. The writing is smooth and the readers can easily follow.
2. Extensive experiments are conducted to verify the effectiveness of the proposed method.

**Weaknesses:**

1. The proposed method essentially achieves knowledge transfer through uploading embeddings, which is not novel. Furthermore, compared to prototype-based methods, it introduces significant additional communication overhead and poses a substantial risk of privacy breaches.
2. To my knowledge, even with heterogeneous models, the experimental results shouldn't be that bad. Cifar10 is a very simple dataset, but its results are still very low with HtFE-1 and a=0.1, so the results are questionable.

**Questions:**

Please see the weaknesses.

---

### Note · Authors · 2025-11-23

**Comment:**

We thank the AC and reviewers for their effort. We are withdrawing our submission to revise and resubmit it to another venue.

**Withdrawal Confirmation:**

I have read and agree with the venue's withdrawal policy on behalf of myself and my co-authors.